# Geochemistry, Mineralogy and Microbiology of Cobalt in Mining-Affected Environments

Gabriel Ziwa [1,2,*], Rich Crane [1,2] and Karen A. Hudson-Edwards [1,2]

1 Environment and Sustainability Institute, University of Exeter, Penryn TR10 9FE, UK; r.crane@exeter.ac.uk (R.C.); k.hudson-edwards@exeter.ac.uk (K.A.H.-E.)
2 Camborne School of Mines, University of Exeter, Penryn TR10 9FE, UK
* Correspondence: gz239@exeter.ac.uk

**Abstract:** Cobalt is recognised by the European Commission as a "Critical Raw Material" due to its irreplaceable functionality in many types of modern technology, combined with its current high-risk status associated with its supply. Despite such importance, there remain major knowledge gaps with regard to the geochemistry, mineralogy, and microbiology of cobalt-bearing environments, particularly those associated with ore deposits and subsequent mining operations. In such environments, high concentrations of Co (up to 34,400 mg/L in mine water, 14,165 mg/kg in tailings, 21,134 mg/kg in soils, and 18,434 mg/kg in stream sediments) have been documented. Co is contained in ore and mine waste in a wide variety of primary (e.g., cobaltite, carrolite, and erythrite) and secondary (e.g., erythrite, heterogenite) minerals. When exposed to low pH conditions, a number of such minerals are known to undergo dissolution, typically forming $Co^{2+}_{(aq)}$. At circumneutral pH, such aqueous Co can then become immobilised by co-precipitation and/or sorption onto Fe and Mn(oxyhydr)oxides. This paper brings together contemporary knowledge on such Co cycling across different mining environments. Further research is required to gain a truly robust understanding of the Co-system in mining-affected environments. Key knowledge gaps include the mechanics and kinetics of secondary Co-bearing mineral environmental transformation, the extent at which such environmental cycling is facilitated by microbial activity, the nature of Co speciation across different Eh-pH conditions, and the environmental and human toxicity of Co.

**Keywords:** cobalt; mine waste; tailings; erythrite; heterogenite; biogeochemistry





## 1. Introduction

Cobalt (Co) is a d-block transition metal and appears in the fourth period of the Periodic Table between iron and nickel [1]. It is a naturally occurring element widely distributed in rocks, soils, sediments, water, plants, and animals [2–4]. Since 2011, Co has been recognised by the European Commission as a "Critical Raw Material" [5] and deemed strategically important [6,7] due to increasing demand and use in rechargeable batteries. Other uses of Co span multiple industries, from healthcare and as a high temperature alloy in combustion engines to renewable energy such as solar and power, and data storage. This is due to its many unique characteristics including magnetic properties and resistance to high temperatures, wear, and corrosion [8].

Cobalt is an essential element necessary for the formation of vitamin B12 (hydroxocobalamin) [2]. However, excessive Co exposure can result in a range of symptoms/conditions in humans including goitre and reduced thyroid activity [9]. One major pathway of Co exposure to humans and plants is from mining activity and associated waste disposal/management. Therefore, a comprehensive understanding of the biogeochemistry of Co in mining-affected environments is crucial in order to achieve sustainable mining practices and remediation of Co contaminated soils and waters. This is important to safeguard against any potential adverse environmental and human health impacts that result from Co exposure [10]. It has become apparent, however, that there are only a limited

number of studies devoted to understanding Co mobility within tailings, soils, smelter waste, and mine waters [10–12]. This paper will review the geochemistry, mineralogy, and microbiology of Co in mining-affected environments. Within this, the major controls on Co uptake and mobility in mine-affected waters, soils, sediments, plants, minerals, and microbes are described, and a synopsis of the key areas for future research are included.

## 2. Geology and Characteristics of Co-Bearing Ore Deposits

The average crustal abundance of Co is approximately 25 ppm, making it one of the least abundant elements [1]. In spite of this, many geological processes have locally concentrated Co to form economically viable deposits. Such deposits are hosted in rocks ranging from Precambrian to Quaternary age and are typically between 0.1 and 0.4 wt. % Co grade. Cobalt is almost always mined as a by-product of copper and nickel [1,13] with the exception of the Bou Azzer deposit in Morocco, where it is the chief commodity [14,15]. According to Slack et al. (2017) [7], 34 Co minerals have been recognised in these deposits [7]. These are predominantly sulphides, arsenides, sulpharsenides, arsenates, cobaltiferous iron sulphides, sulphates, and carbonates. The major Co-bearing ores mined across the globe are cobaltite ($CoAsS$), cattierite ($CoS_2$), carollite (($CuCo)_2S_4$), sphaerocobaltite ($CoCO_3$), cobaltpentlandite [($Co$-$Fe)_9S_8$], siegenite [($Ni,Co)_3S_4$], linnaeite ($Co_3S_4$), smaltite ($CoAs_2$), safflorite [($Co,Fe)As_2$], and skutterudite [($Co,Fe,Ni)As_{2\text{-}3}$] [16–19]. Four principal geological settings host the vast majority of currently economically viable Co deposits (Figure 1): hydrothermal, magmatic, laterites, and chemical precipitate deposits.

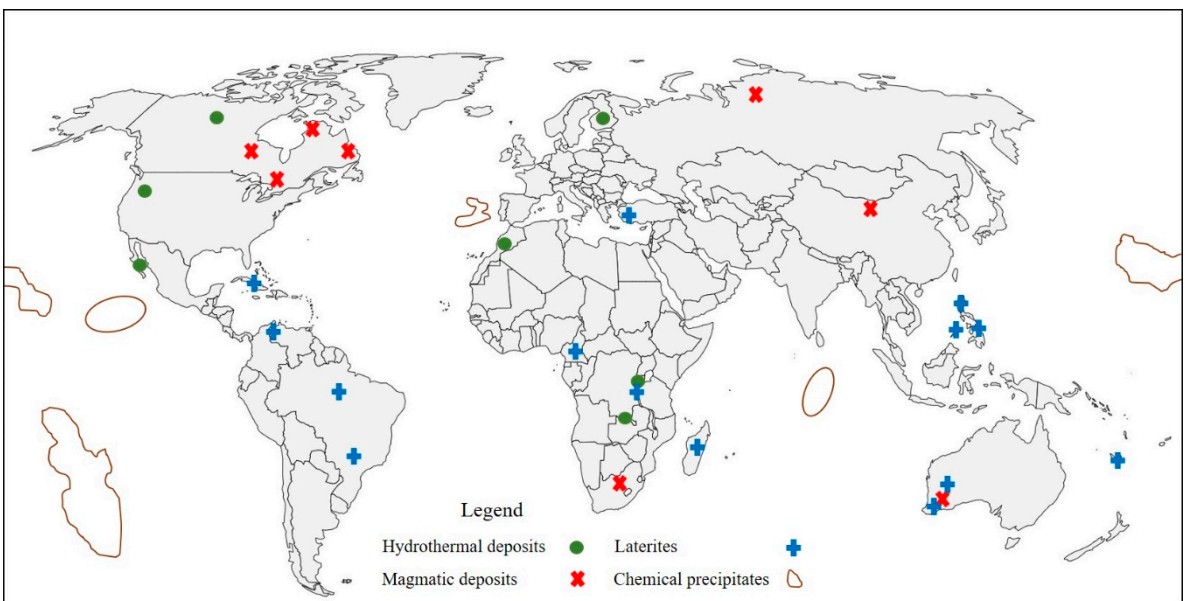

**Figure 1.** Global distribution of different major Co deposits (redrawn from Slack et al., 2017 [7]).

### 2.1. Hydrothermal Deposits

These deposits are formed when hydrothermal fluids interact with basement rocks. These rocks are therefore invariably mafic and/or ultramafic [13] and rich in Fe–Mg [20]. In other cases, the deposits occur within sedimentary basins [21], in which they were most commonly formed by the leaching of Co, Ni, Fe, and As from ultramafic rocks (serpentinites, basalts, peridotites) by acidic magmatic fluids [14,22]. The principle Co-bearing ore minerals of hydrothermal deposits are arsenides, sulpharsenides, and sulphides [23].

### 2.2. Magmatic Deposits

These deposits are produced by high-temperature magmatic processes in some mafic and/or ultramafic intrusions or in volcanic flows [1]. Cobalt is mined as a by-product of

Ni–Cu–PGEs (Platinum-Group Elements) sulphides in mafic-ultramafic intrusions [24]. The concentrations of Ni, Cu, and some recoverable by-product Co sulphides are typically between 0.04 and 0.08 wt. % Co. Slack et al. (2017) [7] showed that ages of the deposits ranged from Archean to Tertiary and are similar to those of their host rocks. The main Co-bearing mineral in these deposits is pentlandite, and linnaeite also occurs in minor amounts [24,25].

### 2.3. Laterites

Laterites are produced by deep humid weathering of bedrock, during which certain elements such as Co and Ni are removed and then enriched by supergene processes. These deposits typically contain 0.1–0.15 wt. % Co and range in age from mid-Tertiary to Quaternary [26–28]. The most significant and/or economic enrichments have been associated with ultramafic rocks [7]. The deposits range in thickness between 10 m and 40 m, and Co is hosted in the clays, goethite, erythrite, and heterogenite [29,30]. In a typical lateritic profile, the highest grades of Ni are found in the saprolite, whereas the highest grades of Co (~0.25 wt. %) are found in the oxidation zones [24,31].

### 2.4. Chemical Precipitates

These are an emerging and potential source of Co in the future [7]. The deposits are associated with Fe and Mn precipitation [7] at the peripheries of seafloor hydrothermal systems, during the formation of Fe–Mn nodules and crusts on the ocean floor and seamounts [32], or during weathering as at Mt. Tabor in Australia [33].

## 3. Geochemistry of Cobalt in Mine Wastes

### 3.1. Cobalt in Mine Waters

A wide range of geochemical conditions have been recorded in mine waters, resulting in variable Co concentrations being reported (Table 1). High Co concentrations in groundwater, adit water, and runoff are mainly attributed to the weathering and dissolution of sulphide ores and secondary minerals in regions of acidic pH. The oxidation of pyrite, and conversely the reduction of Fe, can cause Co and other trace elements to be released from the tailings and other solid mine wastes. According to (Krupka and Serne, 2002) [34], $Co^{2+}$ (aq) tends to be the most dominant aqueous species under geochemical conditions, typically encountered in the natural environment (Figure 2). Studies conducted in Kabwe, central Zambia, where local Pb and Zn ores were mined and processed from 1903 to 1994, have indicated that high concentrations of Co (34,400 mg/L) occur in water flowing from the leach plant at pH 2.89 [35]. At such pH, the oxidation of carrolite ($(CuCo)_2S_4$) can act as a source of dissolved Co in mine waters [35]. High concentrations of Co (0.5–2028 µg/L) also have been recorded in surface waters in Cobalt (Canada) [36]. These have been attributed to the dissolution of secondary minerals such as erythrite and annabergite at pH ~2 [36,37]. According to Zhu et al., (2013) [37], erythrite solubility typically exhibits a strong inverse correlation with solution pH. Cobalt and As dissolution is suggested to proceed via the following reactions [37]:

$$Co_3(AsO_4)_2 \cdot 8H_2O + 2H^+ = 3Co^{2+} + 2HAsO_4^{2-} + 8H_2O \tag{1}$$

$$Co^{2+} + HAsO_4^{2-} = CoAsO_4^- + H^+ \tag{2}$$

$$Co^{2+} + 2OH^- = Co(OH)_2 \tag{3}$$

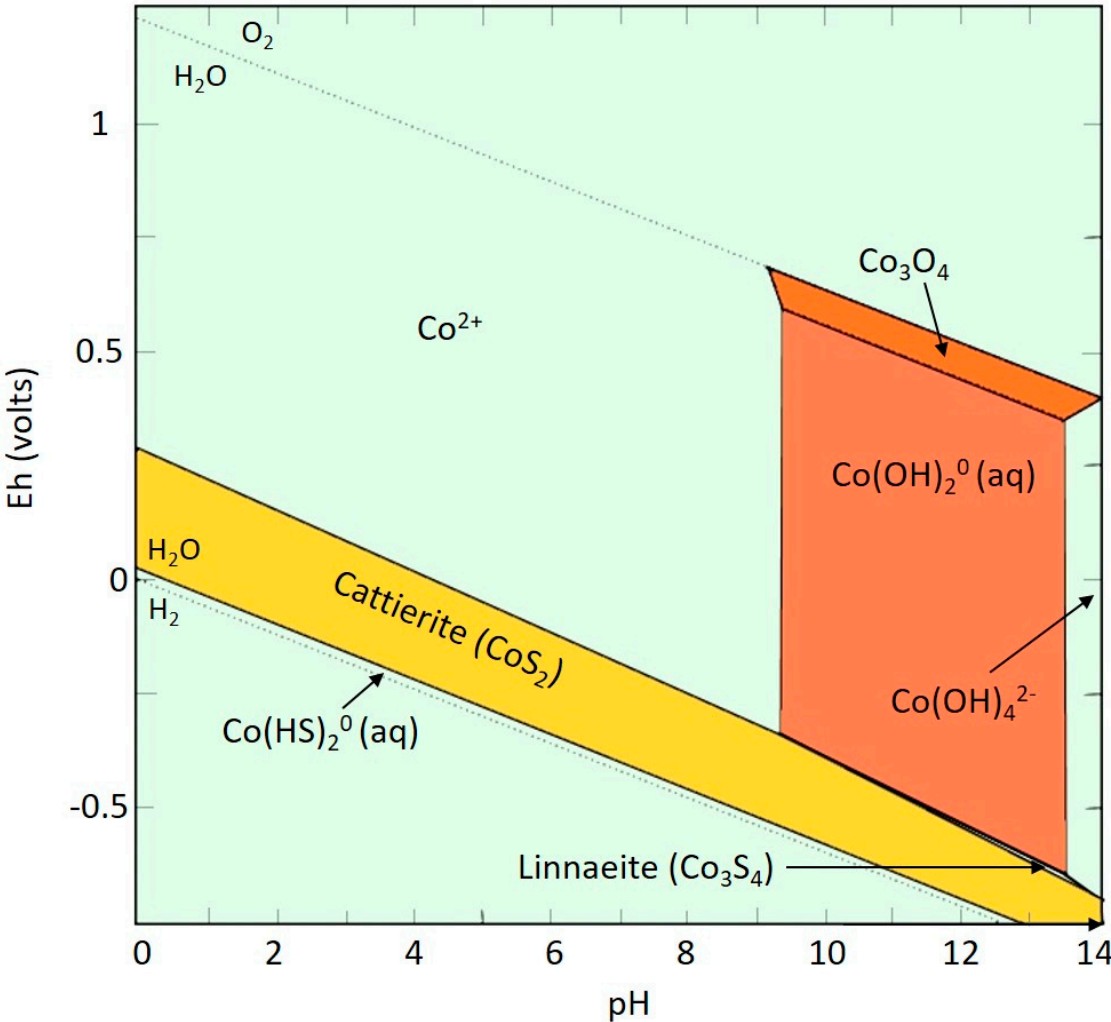

**Figure 2.** Eh–pH diagram showing the dominant Co species (redrawn from Krupka and Serne, 2002 [34]).

In a separate study conducted at the Idaho Cobalt Belt (USA), aqueous Co concentrations were also significantly higher in acidic mine water than those of adjacent circumneutral pH stream waters (e.g., at Blackbird: 75,000 µg/L at pH 2.7, compared to <1.2 µg/L at pH 7.4) [38]. The low pH in the waters proximal to the mines and tailings facilities has been attributed to the weathering of pyrite and pyrrhotite and the subsequent dissolution of cobaltite (CoAsS) [22,38]. In the Central African Copperbelt (Katanga, Democratic Republic of Congo (DRC)), Co concentrations in mining effluent and water (pH ~6) from the Co–Cu deposits have been reported to be as high as 3164 µg/L [39]. The study revealed that the highest concentrations of Co were found in waters close to mining effluent discharge zones [39]. Within this, it has been highlighted that a key mechanism that underpins such Co release to the aqueous phase is heterogenite dissolution, which is particularly prevalent in waters at pH < 6 [34].

Aqueous Co concentrations up to about 18,689 µg/L were reported for acidic (pH 0.6–0.8) and hypersaline leachate seeping from a pyrite pile in the San Telmo mine, Spain [40]. In the same region, Co concentrations in stream waters from the Peña de Hierro abandoned pyrite–Cu mine area (Spain), displayed similar amounts (599–26,100 µg/L in the most acidic (pH 0.7–3.5) streams) [41]. It has been suggested that this high Co concentration comes from Co-bearing pyrite oxidation [41,48].

<div align="center">

**Table 1.** Concentrations of Co in mining-affected waters.

</div>

| Mine/Region | Ore/Deposit Type | Period of Mining | Type | Mean/Range Co Concentration (µg/L) | Reference |
|---|---|---|---|---|---|
| Kabwe mine, Zambia | Pb–Zn | 1903–1994 | Sludge resulting from chemical leaching | 34,400,000 | Sracek et al., 2010 [35] |
| Cobalt, Ontario, Canada | Ag–As–Bi–Co | Not recorded | Ground water samples | 140–1800 | Percival, et al., 1996 [36] |
| Cobalt, Ontario, Canada | Ag–As–Bi–Co | Not recorded | Surface water samples | 0.5–2028 | Percival, et al., 1996 [36] |
| Idaho Cobalt Belt (ICB), USA | Co–Cu–Au | Early 1900s–1967 | Mine water (adits and open pits) | 11,000 | Gray and Eppinger, 2012 [38] |
| San Telmo mine, Spain | pyrite | 1970–1989 | Pyrite leachate, pH 0.61–0.82 | 18,689 | España et al., 2008 [40] |
| Peña de Hierro mine, Spain | Pyrite–Cu | Mid-19th century–1966 | Stream water from the mine | 599–6100 | Romero et al., 2011 [41] |
| Savage River mine, Tasmania, Australia | magnetite | 1967–1982 | Pore waters from old tailings | 5000 | Jackson and Parbhakar-Fox, 2016 [42] |
| Katanga province, DRC | Co–Cu | Before 1960–present | Mining effluent and water | 3164 | Atibu et al., 2013 [39] |
| Rio Piscinas area, Italy | Pb–Zn | beginning of 19th century–1992 | Groundwater samples | 1500–2900 | Concas et al., 2006 [43] |
| Pyrite–uranium mine at Rudki, Poland | pyrite–U | Early 1900–1968 | Acid pool waters from the mine tailings | 303–1439 | Migaszewski et al., 2015 [44] |
| Darrehzar porphyry copper mine, Iran | Cu | Not recorded | Mine water flowing from the mine | 831 | Soltani et al., 2014 [45] |
| Haveri mine, Finland | Au–Cu | 1942–1961 | Ground and surface water | 10–866 | Parviainen, 2009 [46] |
| Banjas area, northern Portugal | As–Au | 1864–1890 | Spring and groundwater proximal to the mine | 11 | Carvalho et al., 2014 [47] |

Acidic Co-bearing mine waters which undergo pH neutralisation have been recorded to result in Co precipitation via a range of reactions including co-precipitation with iron hydroxides and structural substitution onto Fe (oxyhydr)oxides and clay minerals [44,49,50]. For example, groundwater from a legacy Pb–Zn mine in the Rio Piscinas area (Sardinia, Italy) exhibited Co concentrations (2900 µg/L) higher than those of stream waters flowing from the tailings at pH <4.73 (1500–2700 µg/L), whereas even lower values (<1 µg/L) were recorded in distal stream samples (pH 7.67–8.02) [43]. In south-central Poland, acid mine tailings pool waters (pH 2.6–4.0) at a legacy low-grade pyrite–uranium tailings repository had significantly higher concentrations of Co (303–1439 µg/L) than those in adjacent farmers' wells (pH 7.2–8.0; 0.134–0.466 µg/L) [44]. Soltani et al., (2014) [45] reported average Co concentrations of 831.55 µg/L in mine water flowing from the Darrehzar porphyry Cu mine, Kerman province, Iran. Carvalho et al. (2014) [47] tabulated lower Co concentrations of up to 11.91 µg/L in spring and groundwater (pH ~6.1) proximal to the abandoned As–Au mine (Banjas area, Portugal). In SW Finland, at the abandoned Haveri Au–Cu mine, most samples from ground and surface waters had a Co concentration less than 10 µg/L, while some had higher dissolved concentrations of up to 866 µg/L [46]. These variations

between mine site and downstream water were attributed to (a) dilution by surface and groundwater, (b) precipitation or co-precipitation of metallic cations as hydroxides and sulphates; and (c) adsorption of metallic cations by organic and inorganic sediments and aquatic plants [51].

Existing guidelines for Co in irrigation, surface, and ground water are summarized in Table 2. These are relatively limited at present. For example, the World Health Organisation drinking water guidelines for Co are yet to be established [36,52–54]. Co concentrations of the mine waste-affected waters presented exceed guideline values for aquatic life in surface irrigation by many orders of magnitude. Only a few are listed below in the guidelines set for livestock watering.

**Table 2.** Environmental guideline values for Co.

| Type of Limit | Limit Value (mg/kg) | Organisation | Reference |
|---|---|---|---|
| Drinking water | No data | CCME | CCME, 2010 [53] |
| Surface water | 5 | NYSDEC | NYSDEC, 1986 [55] |
| Freshwater for aquatic life | 5 | NYSDEC | NYSDEC, 1986 [55] |
| Agriculture | 50 Irrigation 1000 Livestock | CCME | CCME, 2010 [53] |
| Residential soil quality guidelines | 23 | USEPA | USEPA, 2011 [56] |
| Industrial soil quality guidelines | 300 | USEPA | USEPA, 2011 [56] |
| Sediment Quality Guidelines for the Protection of Aquatic Life | 35 | CCME | CCME, 2010 [53] |

CCME: Canadian Council of Ministers of the Environment
NYSDEC: New York State Department of Environmental Conservation
USEPA: United States Environmental Protection Agency

### 3.2. Cobalt in Tailings and Mine-Affected Soils and Sediments

Concentrations of Co in mine tailings also vary, and selected examples are summarised in Table 3. These variations have been attributed predominantly to the differences in the processing technologies used and to variations in the geology of the ore deposit. For example, in the Central African Copperbelt, Co was recovered historically from Cu flotation concentrates by a Roast–Leach–Electrowin (RLE) process. This technology was ineffective for Co, with recoveries typically from 40 wt. % up to 80 wt. % for oxides and sulphides, respectively [57]. This resulted in substantial amounts of Co being lost to the tailings [58] in dams in the DRC and Zambia (Table 3). Geogenic factors account for the lower Co concentrations reported in Table 3. For example, 57.8 mg/kg Co has been recorded in the tailings resulting from mining Zn–Cu in the Skellefte district sulphide ore field (Sweden) [59]. According to Gavelin (1955) [60], these deposits predominantly contained up to 2 wt. % Zn and 0.001–0.01 wt. % Co. In contrast, higher Co concentrations were found in tailings derived from the mining, processing, and treatment of Co-bearing ores such as the Cu–Co stratiform deposits of DRC and Zambia at 1.9 wt. % Co [16] and 0.5 wt. % Co [17], respectively.

Table 4 summarises the concentrations of Co in mine-affected soils and sediments. Most of these exceed residential soil guideline values (Table 2). According to Pourret et al. (2016) [61], soils accumulate Co due to one or more of the following processes: (i) weathering of soil metal-bearing minerals; (ii) weathering of Co bearing deposits; and (iii) deposition of atmospheric fall-out from ore smelters. The highest concentrations of Co (6.4–21,134 mg/kg) in mining-affected soils occur in the Democratic Republic of Congo (DRC), the largest producer of Co in the world. Narendrula et al. (2012) [62] concluded that due to the mining and processing of Cu and Co, this region is one of the most contaminated mining areas in the world.

**Table 3.** Concentrations of Co in tailings.

| Mine/Region | Ore/Deposit Type | Period of Mining | Tonnage/Type | Mean/Range Co Concentration (mg/kg) | Reference |
|---|---|---|---|---|---|
| Kabwe mine, Zambia | Pb–Zn | 1903–1984 | Oxidised tailings pond | 14,165 | Sracek et al., 2010 [35] |
| Katanga province, DRC | Co–Cu | Before 1960–present | Freshly processed tailings | 6100 | Lutandula and Maloba, 2013 [58] |
| Haveri mine, Finland | Au–Cu | 1942–1961 | 1.5 Mt Oxidised, weathered | 24–329 | Parviainen 2009 [46] |
| Algares area, Portugal | Pb–As sulphides | 1963–1971 | Oxidised zone | 97–157 | Bobos et al., 2006 [63] |
| pyrrhotite mine, Morocco | pyrrhotite | 1964–1981 | >0.4 Mt Oxidised tailings | 60–80 | Hakkou et al., 2008 [64] |
| Azegour mine, Morocco | Cu–Mo–W | 1932–1971 | 850,000 t oxidised tailing impoundments, | 40–440 | Goumih et al., 2013 [65] |
| Skellefte district sulphide ore field, Sweden | Zn–Cu | Not recorded | Freshly processed tailings | 57.8 | Gleisner and Herbert, 2002 [59] |
| The Aljustrel mine (SW Portugal | pyrite | Not recorded | Tailings from roasting pyrite | 59 | Candeias et al., 2011 [66] |
| Virgina Au–pyrite belt, USA | Au–pyrite | 1909–1945 | 120,000 t primary unoxidised and oxidised | 44 | Seal II et al., 2008 [67] |
| Rio Piscinas area, Italy | Pb–Zn | beginning of 19th century–1992 | Not described | 15–43 | Concas et al., 2006 [43] |
| Kidston gold mine, Australia | Au | 1985–2001 | Un-oxidised tailings | 2.32–29.20 | Edraki et al., 2019 [68] |

**Table 4.** Concentrations of Co in mining-affected soils and sediments.

| Mine/Region | Ore/Deposit Type | Period of Mining | Material Type | Mean/Range Co Concentration (mg/kg) | Reference |
|---|---|---|---|---|---|
| Kolwezi district, Province of Lualaba, DRC | Co–Cu | Before 1960–present | Stream sediments | 19.4–18,434 | Atibu et al., 2018 [69] |
| Kolwezi district, Province of Lualaba, DRC | Co–Cu | Before 1960–present | Soil samples | 6.4–21,134 | Atibu et al., 2018 [69] |
| Katanga province, DRC | Co–Cu | Before 1960–present | Stream sediments | 59.7–13,199 | Atibu et al., 2013 [39] |
| Idaho Cobalt Belt (ICB), USA | Co–Cu–Au | Early 1900s–1967 | Stream sediments | 14–520 | Gray and Eppinger, 2012 [38] |
| Idaho Cobalt Belt (ICB), USA | Co–Cu–Au | Early 1900s–1967 | Soil samples | 29–940 | Gray and Eppinger, 2012 [38] |
| Rio Piscinas area, Italy | Pb–Zn | beginning of 19th century–1992 | Stream sediments | 9–38 | Concas et al., 2006 [43] |

**Table 4.** *Cont.*

| Mine/Region | Ore/Deposit Type | Period of Mining | Material Type | Mean/Range Co Concentration (mg/kg) | Reference |
|---|---|---|---|---|---|
| The Kettara Mine, Morocco | Ochre–pyrrhotite | 1933–1982 | Soil samples | 25.14 | El Amari et al., 2014 [70] |
| The Kettara Mine, Morocco | Ochre-pyrrhotite | 1933–1982 | Stream sediments | 27.62 | El Amari et al., 2014 [70] |
| Maldon, Victoria, Australia | Au | 1850s–not reported | Soil samples | 25 | Abraham et al., 2018 [71] |
| Hagan Mine, Egypt Bay, Maine, USA | Cu–Ag | 1877–1885 | Soil samples | 1.9–21.3 | Osher et al., 2006 [72] |
| Panasqueira mine area, Portugal | Sn–W | 1898–2001 | Soil samples | 7–8 | Candeias et al., 2015 [73] |
| Alto da Várzea radium mine, Portugal | Ra–U | 1911–1922 | Stream sediments | 3.8–4.8 | Antunes et al., 2018 [74] |

Such contamination has in part been attributed to extensive exploitation and smuggling of secondary Cu–Co ores by artisanal and unlicensed miners [10] and to a lack of land reclamation programs to address environmental degradation [10]. The few results that do not exceed guidelines include soils from around Hagan Mine, Egypt Bay, Maine, USA (1.9–21.3 mg/kg) [72], and the Panasqueira mine area, Portugal (7–8 mg/kg) [73]. These lower concentrations can be attributed to the fact Co is not enriched in the parent rocks and ore minerals of these regions [72,73]. According to Bradl (2004) [75], adsorption to Fe and Mn (oxyhydr)oxides also plays an important role in metal(loid) retention in soils. Generally, Co metal adsorption is negligible at low pH and then increases at near neutral pH (~5.5) to almost complete adsorption over a relatively small pH range (up to 8) [75]. This phenomena is observed, for example, in the Sudbury region, Canada where Co concentrations in the soils (pH < 4.5) range from 1.6 to 37.9 mg/kg, whereas soils in the DRC region have higher pH (5.7 to 7) with a higher concentration of Co [62].

The sediment quality guideline set by CCME (2010) [53] for the protection of aquatic life is 35 mg/kg Co (Table 2). Cobalt concentrations in stream sediments from the mining regions of DRC [39,69] and the Idaho Cobalt Belt in the USA [38] are many orders of magnitudes higher than this guideline value. Some of the stream sediments from mining regions reported in Table 4 do not exceed this guideline. Studies have shown that concentrations of Co in stream sediment decrease with increasing distance from the mines (Figure 3), suggesting considerable downstream sediment dilution [38,69]. Adsorption onto river beds and accumulation into the hyporheic zone as a function of low pH [76] have also been used to explain downstream decreases in Co concentrations in stream sediments. Fuller and Harvey (2000) [77] demonstrated that about 52% Co was removed from stream sediments into the hyporheic zone by sorption to manganese oxides.

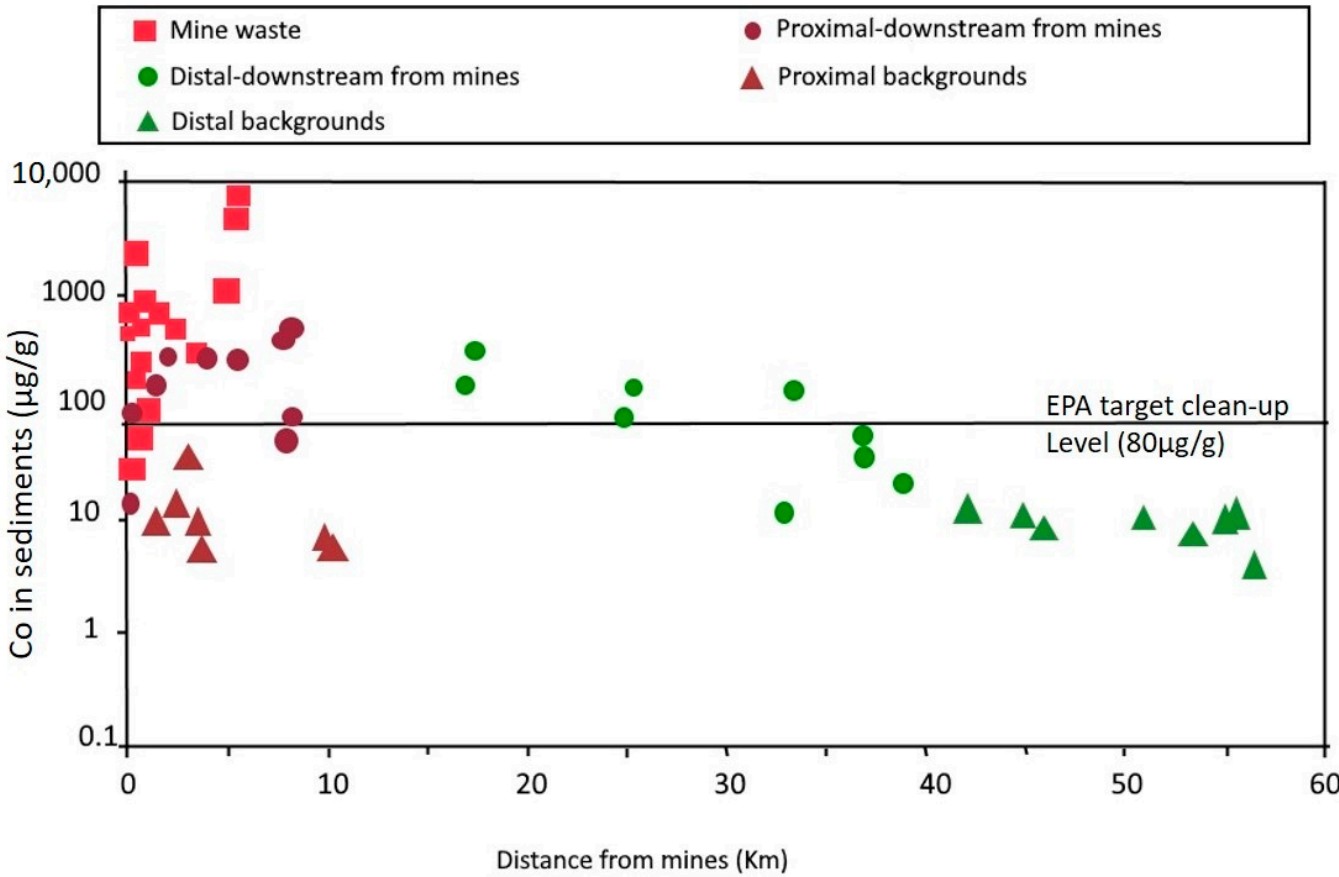

**Figure 3.** Stream sediment concentration of Co decreasing with increasing distance from mines in the ICB (redrawn after Gray and Eppinger, 2012 [38]).

### 3.3. Cobalt in Mine-Affected Plants

Very few studies have reported concentrations of Co in mining-affected plants. Some existing data are presented in Table 5. A study in the DRC showed that concentrations of Co as high as 5050 mg/kg had accumulated in the leaves of the plant *Phalaris arundinacea* L. Such plants are also known to thrive in pH ranges between 7.3 and 8.8 and to survive persistent anoxic phases [78]. As such, these have been suggested to be potentially useful for the remediation of Co contaminated soils [69]. Their relatively shallow root systems (often <60 cm depth), however, dictate that such technology may only be useful for near-surface Co extraction (Comes, 1971) [79]. These plants have been classified as hyperaccumulators because the dry weight Co concentration in their tissue exceeds 1000 mg/kg (Zayed et al., 1998) [80]. Other Co hyperaccumulators have been reported including *Crotalaria cobalticola*, *Crassula vaginata,* and *Haumaniatrum robertii* (Bakkaus et al., 2005) [81].

In a separate study, Křı́bek et al. (2014) [82] compared the concentration of Co in Cassava leaves grown on contaminated and uncontaminated soils in the Copperbelt, Zambia. The study showed that concentrations of Co were up to three times higher in the plants grown on contaminated soils. In Morocco, El Hamiani et al. (2015) [83] noted that Co concentrations were also higher in plants growing on soils within the vicinity of an old Co–Ni mine than those growing in the vicinity of an Mn mine and a Cu mine. According to Luo et al. (2010) [84], the soil-to-plant transfer factor for Co was also higher for the Co–Ni mine and in certain plant species (*Vicia faba* and *Rosmarinus officinalis*), exceeding the Agency for Toxic Substances and Disease Registry (ATSDR) range of 0.01–0.3 [2].

## 4. Mineralogy of Cobalt in Mine Wastes

Co-bearing minerals most commonly detected in mine waste are summarised in Table 6. Amongst the most frequently identified is the sulpharsenide cobaltite. Cobaltite is one of the major minerals found in the wastes of epigenetic Cu–Co–Ni deposits [85,86], the five-element (Ag–Bi–Co–Ni–As) vein deposits [87,88] and many massive sulphide deposits such as the Lousal Mine in the Iberian Pyrite Belt, Portugal [89], in Cu–Au–Ag deposits of greenstone belts [90], and in Au–As–Cu polymetallic deposits of Poland [91]. Soils affected by mine activities in Spain were found to contain small cobaltite crystals (<10 μm) at concentrations of up to 1.8% Co [86]. Using QEMSCAN® with energy dispersive spectrometer (SEM-EDS) analysis, Rollinson et al. (2018) [92] documented primary cobaltite (0.41–50.05 volume/area %) and a secondary phase, erythrite, up to 5.9 volume/area % on the northeastern coast of Cornwall, UK. The cobaltite was mainly associated with chalcopyrite and erythrite [92]. Other Co-bearing minerals found in mine wastes elsewhere include primary arsenides safflorite and skutterudite [93]. In mine tailings in Cobalt, Canada, Clarke (2017) [93] observed these mineral phases as fine grained aggregates.

At low pH, Co-bearing arsenides and sulpharsenides undergo oxidative dissolution, releasing aqueous Co into the environment. When the pH is increased to near neutral by the presence of, for example, carbonates, the dissolved Co reacts with soluble As oxyanions to form the secondary arsenate phase erythrite [93,94]. Clarke (2017) [93] observed this phase as precipitation rims on silicates and carbonates in a sample taken from high grade tailings from Cobalt, Ontario, Canada. The oxidation of cobaltite resulting in the formation of secondary erythrite has been suggested to follow the reaction steps summarised in Equations (4) and (5) [94].

$$4CoAsS + 13O_2 + 6H_2O = 4CoSO_4 + 4H_3AsO_4 \tag{4}$$

$$3CoSO_4 + 2H_3AsO_4 + 8H_2O = Co_3(AsO_4)_2 \cdot 8H_2O \text{ (erythrite)} + 3H_2SO_4 \tag{5}$$

Markl et al. (2014) [95] demonstrated, using PHREEQC modelling, the precipitation of erythrite from dissolution of Co arsenide phases (safflorite-skutterudite) and concluded that this reaction could scavenge Co from natural solutions. This can be summarised in a two-step reaction as follows [95]:

$$CoAs_2 \text{ (safflorite)} + 2H_2O + 3O_2 = Co^{2+} + 2H_2AsO_4^- \tag{6}$$

$$3Co^{2+} + 2H_2AsO_4^- + 8H_2O = Co_3(AsO_4)_3.8H_2O \text{ (erythrite)} + 4H^+ \tag{7}$$

A number of studies have reported that Co precipitates as a trace element in other mineral phases such as Fe(III) oxyhydroxides and Cu sulphides (Table 7) in acid oxidising mine wastes. Sracek et al. (2010b) [96] demonstrated that Fe(III) oxyhydroxides in tailings contained up to 1.89 wt. % Co. With the aid of X-ray diffraction, Sracek et al. (2010a) [35] studied precipitated efflorescent salts resulting from chemical leaching of Cu–Co concentrates.

It was shown that Co was present in bloedite (2.28 wt. %) and also in moorhouseite ($CoSO_4 \cdot H_2O$), but the concentration of Co in the latter was not reported. In a similar study in the DRC, crusts of the pinkish efflorescent salt hexahydrite ($MgSO_4 \cdot 6H_2O$) had high Co/(Co + Mg) values up to 25 at. % [104]. Between 0.005 and 0.03 wt. % Co has also been recorded in Fe oxyhydroxides in sulphide tailings at Stekenjokk in northern Sweden [105]. Trace amounts of Co (0.030, 0.121, and 0.175 wt. %) were detected in bornite, chalcocite, and covellite, respectively, in Texeo mine waste in Spain [85]. About 0.66 wt. % Co was also detected in the grains of Fe metal, 0.83 wt. % Co in pyrite and 0.08 wt. % Co in sphalerite in the slag at Hopewell mine, Pennsylvania, USA [111].

**Table 5.** Concentrations of Co in mining-affected plants.

| Mine/Region | Ore/Deposit Type | Period of Mining | Type | Mean/Range Co Concentration (mg/kg) | Reference |
|---|---|---|---|---|---|
| Kolwezi district, Province of Lualaba, DRC | Co–Cu | Before 1960–present | *Phalaris arundinacea* L. | 9–5050.80 | Atibu et al., 2018 [69] |
| Copperbelt Province, Zambia | Co–Cu | Before 1960–present | Cassava leaves *(Manihot esculenta crantz)* | 24 | Kříbek et al., 2014 [82] |
| Co-Ni-mine, Southern Morocco | Co–Ni | Not reported | Parsley *(Petroselinum vulgare)* | 20.2–69.4 | El Hamiani et al., 2015 [83] |
| | | | Rosemary *(Rosmarinus officinalis)* | 39.1–54.4 | |
| | | | Fava bean *(Vicia faba)* | 74.6 | |
| Ishiagu, South East Nigeria | Pb–Zn | Not reported | Roots *(Clotalariaretusa* and *Andropogontectorum)* | 13.40–89.75 | Ogbonna et al., 2015 [97] |
| | | | Stems *(Imperatacylindrica* and *Alchorneacordifolia)* | 2.20–78.20 | |
| Palão and Pinheiro mines, Portugal | Pb–Zn | Not reported | *Elatine macropoda* | 127.8 | Prasad et al., 2006 [98] |
| Shangla District, Pakistan | Cr | Not reported | Roots *(N. cataria)* | 23 | Nawab et al., 2015 [99] |
| Sukinda chromite mine, India | Cr | Not reported | *Solanum surattense* | 9.9 | Samantaray et al., 2001 [100] |

**Table 6.** Co-bearing minerals in mine wastes.

| Mineral Name | Elemental Composition | References |
|---|---|---|
| Cobaltite | $(Co,Fe)AsS$ | Harris et al., 2003 [101]; Kelly et al., 2007 [87]; Percival et al., 2007 [88]; Loredo et al., 2008 [85] |
| Carrollite | $CoCu_2S_4$ | Chen et al., 2016 [102] |
| Sphaerocobaltite | $CoCO_3$ | Vítková et al., 2010 [103] |
| Cobaltpentlandite | $(Co-Fe)_9S_8$ | Vítková et al., 2010 [103] |
| Safflorite | $(Co,Fe,Ni)As_2$ | Clarke, 2017 [93] |
| Skutterudite | $(Co,Ni,Fe)As_{3-x}$ | Clarke, 2017 [93] |
| Erythrite | $Co_3(AsO_4)_2 \cdot 8H_2O$ | Percival et al., 2007 [88]; Loredo et al., 2008 [85]; Clarke, 2017 [93] |
| Bieberite | $CoSO_4 \cdot 7H_2O$ | Sracek et al., 2010 [35]; Mees et al., 2013 [104] |
| Moorhouseite | $CoSO_4 \cdot H_2O$ | Sracek et al., 2010 [35] |

**Table 7.** Common minerals in mine waste containing Co as a trace element.

| Mineral Name | Elemental Composition | References |
|---|---|---|
| Fe oxyhydroxides | FeOOH | Holmström and Öhlander, 2001 [105]; Sracek et al., 2010 [96]; Queiroz et al., 2018 [106] |
| Pyrite | $FeS_2$ | Moncur et al., 2005 [107]; Jackson and Parbhakar-Fox, 2016 [42]; Zhang et al., 2020 [108] |
| Arsenopyrite | FeAsS | Assawincharoenkij et al., 2018 [109] |
| Pyrrhotite | $Fe_{(1-x)}S$ | Moncur et al., 2005 [107]; Heikkinen and Räisänen, 2008 [110] |
| Co-poor bloedite | $Na_2(Co,Mg)(SO_4)_2 \cdot 4H_2O$ | Sracek et al., 2010 [35] |
| Bornite | $Cu_5FeS_4$ | Loredo et al., 2008 [85] |
| Chalcocite | $Cu_2S$ | Loredo et al., 2008 [85] |
| Covellite | CuS | Loredo et al., 2008 [85] |
| Chalcopyrite | $CuFeS_2$ | Assawincharoenkij et al., 2018 [109] |

## 5. Microbiology of Cobalt in Mine Wastes

Due to the limited sources and supply of, and the increasing demand for Co [112], there is a significant global interest in applying bioleaching techniques to recover Co from mine waste [113]. This is because conventional pyrometallurgy and hydrometallurgy methods can, under certain circumstances, be less efficient for Co recovery, and as such, require greater energy input, ongoing management, and/or leave behind a legacy of waste [114,115]. Microorganisms capable of oxidizing Fe or S have been identified in tailings and other sulphide-bearing mine wastes [116]. The Fe- and S-oxidising bacteria *Thiobacillus (T.) ferrooxidans*, *Acidithiobacillus (A.) ferrooxidans*, and *A. thiooxidans* have been studied for their ability to oxidise sulphide ores that in turn results in substantial acid generation [113,114,117]. These bacteria do not oxidise Co sulphide ore minerals (CoAsS, $Co_9S_8$, $CoS_2$, (Co,Fe)AsS, etc.) directly. Instead, they gain energy by oxidizing the $S^{2-}$, $S^0$, $S_2O_3^{2-}$, and other transitional S species of the minerals and thereby liberate Co in the process [117].

In a study to determine whether *T. ferrooxidans* could dissolve Co from sulphidic smelter waste, it was noted that the reaction rate increased after adding pyrite to the fine grained <270 μm material. It was concluded that substantial (up to 70% recovery) amounts of Co can be released from sulfidic mine waste [113]. Coto et al. (2007) [114] used *A. thiooxidans* to leach Co from laterite tailings containing about 890 mg/kg Co. The tailings were waste resulting from the extraction and processing of lateritic Co and Ni. The *A. thiooxidans* cell were inoculated in 0 K medium with an initial pH of 3.0 and 2% *w/v* S as the source of energy. After 48 h, the pH decreased to 1 due to the acidity produced by the bacteria. By adding tailings to a pulp density of 2.5% *w/v*, 80% of their included Co was extracted after 15 d. Cabrera et al. (2011) [118] built on this work by extending the time and increasing the pulp density to >10% *w/v*. A recovery of 86% and 89% Co was obtained after 65 and 83 d, respectively.

A number of studies have been conducted on Co using a mixed culture of Fe- and S-oxidising microorganisms. For example, Ahmadi et al. (2015) [119] used a mixed culture consisting of *Leptospirillum ferriphilum*, *Acidithiobacillus caldus*, *Sulfobacillus*, sp. and *Ferroplasma* sp. to recover Co from low-grade Cu–Co–Ni bearing sulfidic tailings (0.044 wt. % Co) from Kerman Province, Iran. In this experiment, about 59.5% of the Co was extracted through the bioleaching process using a pulp density of 5% (*w/v*), pH 1.8, and at a temperature of 45 °C. In another study in Germany, Zhang et al. (2020) [109] used a microbial consortium of *At. thiooxidans*, *At. ferrooxidans*, *Leptospirillum ferrooxidans*, and *Ferroplasma acidiphilum* to extract 91% Co from the Bollrich tailings pond (0.02 wt. % Co).

The study also observed that Co occurred on the surface of framboidal pyrite and was leached by microbial attack.

Overall, optimum conditions for bioleaching of Co from sulphide ores are proposed to be: growth temperature of 35–46 °C, solids concentration (pulp density) 10–15 wt. %, particle size <65 μm, and pH between 1.3 and 2.0 [120].

Other studies have investigated Co and Ni dissolution from laterites and pyritic ores by use of specific fungi species [121–123]. Such processes are currently considered cost-effective due to the fact that fungi can often be grown cheaply and with limited environmental impact, however, studies that have quantified this remain limited [124,125]. In a study to establish whether fungi could be used for bioleaching of metals from mine tailings, Ilyas et al. (2013) [126] observed that 60% Co was solubilised after 24 d at pH (5–7.9). Similarly, Newsome et al. (2020) [127] used the fungal community to recover Co from laterites and recorded that up to 64% Co could be recovered via microbial reduction of Mn(IV)- to Mn(II)-oxides, releasing Co(III) from the crystal structure.

## 6. Geochemical-Mineralogical-Microbiological Controls on Cobalt Mobility in Mining-Affected Environments

The geochemical behaviour of Co is generally similar to that of Fe and Mn, and its concentration in mine affected waters, stream sediments, and soil is primarily controlled by adsorption and co-precipitation reactions with Mn and Fe (oxyhydr)oxide minerals [128]. Cobalt aqueous geochemistry is dominated by +2 and +3 oxidation states [13], with $Co^{3+}$ being thermodynamically unstable and changing under Eh-pH conditions prevalent in most natural waters. However, the presence of certain complexing ligands such as EDTA and $NH_3$ can stabilize $Co^{3+}$ and allow it to persist in aqueous solutions [34]. Figure 4 summarises the geochemical and mineralogical relationships between these aqueous Co species and the various Co-bearing minerals found in mine wastes.

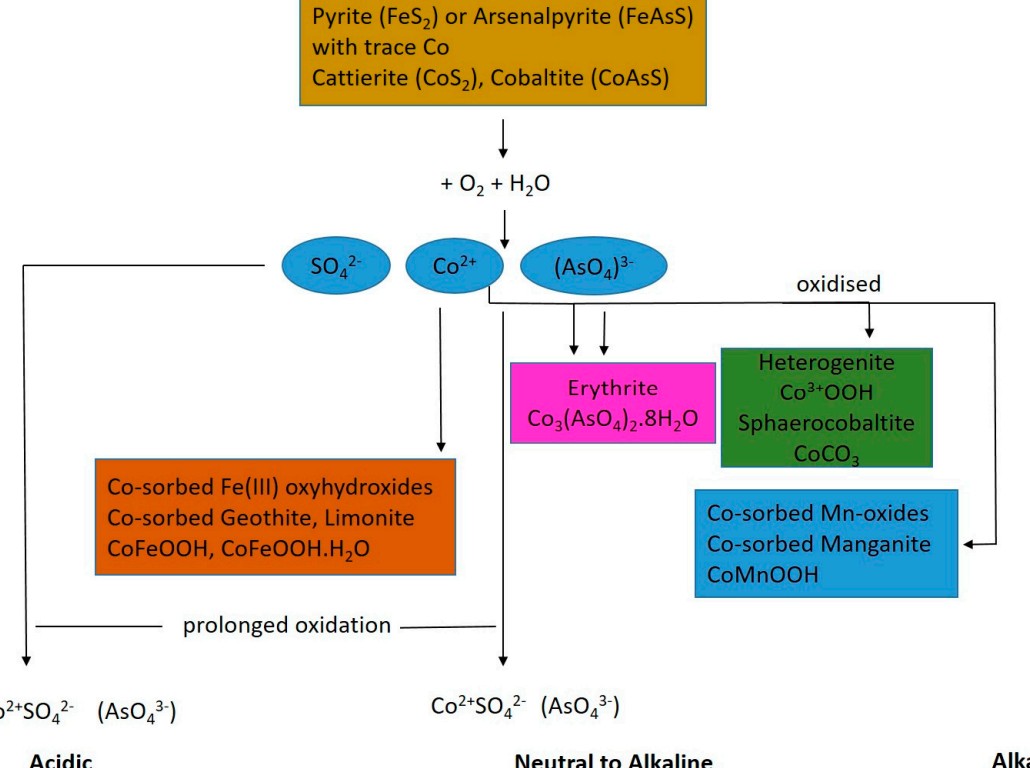

**Figure 4.** A schematic illustration of the key mechanisms that govern Co dissolution and secondary mineral precipitation from mine waste as a function of pH. Rectangular and oval boxes correspond to solid minerals and aqueous species, respectively.

### 6.1. Impact of Eh-pH on Co Geochemistry

The dominant Co-bearing minerals expected to be present in mine waste under reducing conditions and over a wide pH range are cattierite and cobaltite [34,93]. At low pH (<4) the speciation of cobalt is dominated by $Co^{2+}$ and $CoSO_4^0$ [96]. At near neutral to alkaline pH and under oxidising conditions, the $Co^{2+}$ will react with As oxyanions to produce secondary erythrite in an environment saturated with As oxyanions [93]. However, when oxidation is prolonged at pH ~6, Co may be remobilised in the mine water [94]. When the solution pH increases to >7, $Co^{2+}$ can also be rapidly polymerized, leading to the formation of colloidal $Co(OH)_2$, which can be readily oxidised by aqueous oxygen, as in reactions (8) and (9) [129];

$$Co^{2+} + 2H_2O = Co(OH)_{2(c)} + 2H^+ \tag{8}$$

$$Co(OH)_{2(c)} + \frac{1}{4}O_{2(aq)} = CoOOH_{(c)} + \frac{1}{2}H_2O \tag{9}$$

The remobilized Co in surface waters re-precipitates as heterogenite (CoOOH) as soon as the environment becomes less oxidizing and more alkaline [130,131]. In alkaline pH environments, $Co^{2+}$ is oxidised to, and adsorbed on, Mn(IV) oxides as less soluble $Co^{3+}$, as in Equation (10) [132,133].

$$2Co^{2+}_{(aq)} + 2MnO_{2(s)} + 3H_2O_{(l)} \rightarrow 2CoOOH_{(s)} + Mn_2O_{3(s)} + 4H^+_{(aq)} \tag{10}$$

Relatively insoluble $Co^{3+}$, for example, in heterogenite, can be solubilised as $Co^{2+}$ by reductive dissolution in the presence of reducing agents such as sulphur dioxide ($SO_2$) and ferrous ions ($Fe^{2+}$) under reducing and acidic conditions [134]. Reduction of $Co^{3+}$ to $Co^{2+}$ by sulphite is represented by the Equation (11) [134].

$$SO_3^{2-}_{(aq)} + 2Co^{3+}_{(s)} + H_2O_{(l)} \rightarrow SO_4^{2-}_{(aq)} + 2Co^{2+}_{(aq)} + 2H^+_{(aq)} \tag{11}$$

Adsorption on, or co-precipitation with Fe(III) and Mn(IV) (oxyhydr)oxides, as a function of pH, metal concentration, and temperature are other important processes influencing Co behaviour in the environment [96,130]. Cobalt is known to be adsorbed onto secondary Fe(III) (oxyhydr)oxides that are characteristic of mining-affected environments under neutral to moderately acidic pH conditions [135]. In the presence of Fe(III), and at pH between 5.5 and 8, $Co^{2+}$ will be depleted from mine waters and adsorbed onto minerals such as magnetite ($Fe_3O_4$), hematite ($Fe_2O_3$), and goethite ($\alpha FeOOH$) [136,137].

According to Hem et al. (1985) [129], reactions involving Mn promote the oxidation of Co. $Co^{2+}$(aq) reacts with Mn oxides (Equation (10)) to precipitate heterogenite and release $Mn^{2+}$(aq) [130]. Mn-oxide colloids scavenge Co via adsorption and/or co-precipitation reactions in which hydrated Co cations are attracted to the negatively charged surfaces of Mn-oxides like manganite (MnOOH), birnessite ($\delta$-$MnO_2$) [44,131].

### 6.2. Impact of Microbial Activity

Microbial interactions are widely regarded as exhibiting a central role in controlling Co environmental mobility [127]. Co mobilisation in mine waste rocks and minerals, tailings, soils, and stream sediment can occur via a number of processes such as redox processes, protonolysis, complexation by excreted metabolites and Fe(III)-binding siderophores, and indirect Fe(III) attack [138]. Microbial activity reduces Fe(III) and Mn(IV) to Fe(II) and Mn(II), respectively [139]. This results in an increase in solubility and consequently, release of the Co adsorbed to Fe(III) and Mn(IV) oxides [127,132,138,139]. Bacteria such as *Acidithiobacillus ferrooxidans, Leptospirillum ferrooxidans, Sulfolobus spp.*, and *Acidianus brierleyi* can also oxidise Co-bearing sulphides [138,140]. This microbial activity can cause the release of Co by either direct oxidative attack (of the crystal lattice of the Co-bearing sulphides) or indirect oxidative attack by generating acid ferric sulphate, which then oxidizes the Co-bearing sulphide [138,140].

Biosorption processes also play an important role in controlling Co mobility in mining affected environments [141]. Depending on the pH conditions, Co can sorb to biosorbents such as algae, fungi, and bacteria via a number of mechanisms including electrostatic (physical) adsorption, ion exchange, precipitation, and co-precipitation [125,142]. In low pH environments, biosorption of Co is usually low. This is due to increased protonation, which results in the repulsion between the metal ions ($Co^{2+}$) and the functional group (carboxyl, hydroxyl) at the binding sites of the biosorbent [141]. Increasing the pH deprotonates the functional groups, leaving them excited to attract Co ions, thereby increasing the rate and biosorption capacity [143].

## 7. Conclusions

In this review, Co geochemistry, mineralogy, and microbiology in mine-affected environments were summarised. Many studies have provided information on Co concentrations and geochemical behaviour across mine waters, tailings, stream sediments, soils, and plant environments. There remains a lack of comprehensive data, however, and as such, a reliable understanding of Co distribution across different mining environments is yet to be established. A key area within this is a current lack of Co speciation data, particularly across the entire breadth of Eh-pH conditions commonly encountered within mining environments, and a subsequent lack of kinetic and mechanistic understanding of the specific hydrogeochemical conditions under which secondary Co-bearing mineral precipitates form including their stability thresholds. There is also a lack of Co geochemistry data from the Central African Copperbelt, which is amongst the most Co-contaminated mining environments in the world. This is therefore likely the result of the implementation of imperfect Co mining and/or waste disposal techniques with resultant environmental and human health damage. Another area of major knowledge gap is a lack of regulatory framework for Co in waste and permitted environmental discharge. For example, the WHO drinking water guidelines for Co are yet to be established. Finally, the role of bacteria and other microorganisms in the cycling of Co in different mining environments has recently received some attention, but the specific conditions and mechanisms under which these microbes reduce Co(III) to Co(II) remain relatively poorly understood. Overall, it is argued that further research is urgently required on both fundamental Co hydrogeochemistry and geomicrobiology, but also more targeted studies that link such data to the establishment of new regulatory frameworks and management/remediation policies across different mining environments are needed. This will help limit the likely increase in environmental and human health damage as Co mining activities continue to expand this century.

**Author Contributions:** G.Z. collected the data; G.Z., R.C. and K.A.H.-E. formulated and wrote the review paper. All authors have read and agreed to the published version of the manuscript.

**Funding:** This work was supported by the UK Foreign, Commonwealth & Development Office (FCDO) through the Commonwealth Scholarship Commission, grant number ZMCS-2018-855.

**Data Availability Statement:** Data sharing not applicable. No new data were created or analyzed in this study. Data sharing is not applicable to this article.

**Acknowledgments:** We would like to thank the two anonymous reviewers for their guidance throughout the processes of developing this manuscript.

**Conflicts of Interest:** The authors declare no conflict of interest.

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
