# Peer review of "Geochemistry, Mineralogy and Microbiology of Cobalt in Mining-Affected Environments"

_minerals, doi:10.3390/min11010022_

Round 1

Reviewer 1 Report

Review of Manuscript “Geochemistry, Mineralogy and Microbiology of

Cobalt in Mining-Affected Environments”, Submitted to Minerals by Gabriel Ziwa, Rich Crane, and Karen A. Hudson-Edwards

Summary

The manuscript presents a comprehensive review of key geochemical, mineralogical and microbiological considerations for cobalt (Co)-rich environments, particularly pertaining to ore and mine waste deposits. The reviewer appreciates this effort and agrees that more work should be done to fill gaps related to the biogeochemical effects of mine waste on the global environment. The article had a sufficient amount of citations for a review article, from several study areas across the world. Different aqueous chemistry modeling and field scale techniques were represented in works that were summarized in the article. While the characterization/methods in the citations are generally adequate, the scope of the studies included seem narrow. For instance, the authors discuss Co microbiology mainly in the context of bioleaching and smelter waste. Overall, the microbiology section could be strengthened. The manuscript would benefit from a thorough English grammar revision. I have highlighted some issues in the specific comments below. As presented in the draft submitted, I recommend major revisions (including the authors’ complete omission of Figure 4) before consideration for publication.

Specific Comments

  • First, it was difficult to reference a page and line number for providing comments since the page numbers are not consistent, and subsequent line numbers can be repeated. It would be helpful to have these in order before sending for review.
  • Line 44. The following statement does not make sense and should be reworded. “This is important to safeguard against any potential adverse environmental and human health impacts that from Co exposure [10].”
  • Line 47-49. Authors should consider removing ‘animals’ from the statement, since animals are not discussed again thereafter, “Within this, the major controls on Co uptake and mobility in mine-affected waters, soils, sediments, plants, animals, minerals and microbes are described, and a synopsis of the key areas for future research are included.” If a discussion and literature on speciation and bioavailability for humans/animals were included, this would be appropriate.
  • Line 61. ‘Principal’ is the correct version to use in this statement and in others that wish to describe a sense of importance or high prominence. “Four principle geological settings host the vast majority of currently economically viable Co deposits (Fig. 1):”
  • Line 123: Reword this statement. As it reads, does not make sense. “At pH <6 the within mine effluents and surface waters in equilibrium with atmospheric CO2, heterogenite is dissolved and the metal mobilized as aqueous Co2+(aq)[34].”
  • Line 162: Again, revise, “For example, For example, in the Central African Copperbelt, Co was recovered historically from Cu flotation concentrates by a Roast-Leach-Electrowin (RLE) process.”
  • It would be helpful to include somewhere a reference of how much waste is generated during extractive practices. The authors alluded to this a little when pointing out that the Democratic Republic of Congo is a significant source of global Co. A few hundred tons generated vs. several hundreds of millions of tons can have different implications for mobility and recovery.
  • Table 1 and Table 4 have an adequate amount of literature referenced for Co concentrations in waters and sediments/soils; Can the authors include more than four references in Table 5 related to plants? Plants are an increasingly explored area of study and would add value to the review article. Are there known plants that are hyperaccumulators of Co? That are tolerant? If not, make a note in the conclusions that highlight this as a research gap.
  • Section 5. Microbiology of Cobalt in Mine Wastes, discusses microbiology mainly in the context of bioleaching and smelter waste. This section could be strengthened by generally discussing the biotic mechanisms responsible for mobility, speciation, and bioavailability from mine wastes. In this context, the authors should consider adding biosorption and possible biotransformation of Co by fungi, which are commonly overlooked microorganisms to bacteria. To help connect the sediment and plant sections, there is an opportunity here to highlight rhizosphere and redox processes from a mechanistic perspective for the uptake or accumulation of Co (ingestion is a major exposure pathway if humans and animals are in contact with such plants). Also, what specific microorganisms have been identified from mine waste sites highlighted in several of your tables where Co is measured in water, sediment, plants?
  • The use of the word ‘deep’ is a bit unclear in this sentence to describe oxidation. Do the authors mean at the surface, related to depth, or for a long period of time? “However, when oxidation is deep and prolonged at pH~6, Co may be remobilised in the mine water [93].”
  • In section 6.1. Impact of Eh-pH on Co geochemistry, are the authors aware of any reductive mechanisms for Co?
  • Figure 4, which summarizes the geochemical and mineralogical relationships between aqueous Co species and the various Co-bearing minerals found in mine wastes is completely missing from the manuscript. I cannot say for sure if it is a valuable addition to the manuscript.

Reviewer 2 Report

The authors of this manuscript provided a thorough review of the knowledge on cobalt in mining-affected environments worldwide. They described the geology and characteristics of Co-bearing ore deposits, current situations regarding Co in mining wastes and its environmental burden, and geochemical, mineralogical, and microbiological behaviors of Co in mine wastes. The authors also summarized the problems in treatment of Co during mining waste disposals, as well as expected directions for future works.

Review comments:

This paper is a very good and useful review of the current situations regarding Co in mining wastes. The structure of the manuscript seems to be well-organized and discussed from a wide perspective. I recommend publication of this manuscript after considering several minor points described below.

[Chapter 2] As a thorough review paper, it will be better to show some representative deposits currently producing Co for each type of deposit described. If possible, please also review the data of their productions, reserves, and resources of Co for each type of deposit, which will make this manuscript further valuable.

[Line 82] “0.1-0.15%” Is it “wt.%”?

[Line 121-122] “This study” Which study does it mean? Ref. 39 (Atibu et al., 2013)? Or the authors’ current manuscript? Please clarify.

[Table 2] Please provide the full spelling of the organization names.

[Line 162] Please delete one of the duplicated “For example”.

[Line 306-307] Were the adsorption of Co on framboidal pyrite and their leaching observed by Zhang et al. (2020)? Or by another research? Please clarify.

[Line 319] I cannot find Figure 4. Please check.
